# Microtubule-Associated Serine/Threonine (MAST) Kinases in Development and Disease

**DOI:** 10.3390/ijms241511913

**Published:** 2023-07-25

**Authors:** Marie Rumpf, Sabine Pautz, Benedikt Drebes, Friedrich W. Herberg, Hans-Arno J. Müller

**Affiliations:** 1Department of Developmental Genetics, Institute of Biology, University of Kassel, 34321 Kassel, Germany; rumpf.marie@uni-kassel.de (M.R.);; 2Department of Biochemistry, Institute of Biology, University of Kassel, 34321 Kassel, Germany

**Keywords:** protein phosphorylation, MAST kinase, cell signaling

## Abstract

Microtubule-Associated Serine/Threonine (MAST) kinases represent an evolutionary conserved branch of the AGC protein kinase superfamily in the kinome. Since the discovery of the founding member, MAST2, in 1993, three additional family members have been identified in mammals and found to be broadly expressed across various tissues, including the brain, heart, lung, liver, intestine and kidney. The study of MAST kinases is highly relevant for unraveling the molecular basis of a wide range of different human diseases, including breast and liver cancer, myeloma, inflammatory bowel disease, cystic fibrosis and various neuronal disorders. Despite several reports on potential substrates and binding partners of MAST kinases, the molecular mechanisms that would explain their involvement in human diseases remain rather obscure. This review will summarize data on the structure, biochemistry and cell and molecular biology of MAST kinases in the context of biomedical research as well as organismal model systems in order to provide a current profile of this field.

## 1. Introduction

Protein kinases belong to the second-largest protein superfamilies encoded by eukaryotic genomes. The human genome encodes about 518 kinases, roughly half of which are mapped to loci associated with genetically predisposed diseases and over 200 of which are identified as mutated in cancer by the ‘Mutations of Kinases in Cancer’ (MoKCa) database [1,2]. Protein kinases are key mediators in cellular signaling pathways and many other cellular processes. Their function relies upon the transfer of a phosphoryl group onto proteins to influence the structure, activity, stability, and/or localization of the respective substrates. The catalytic activity of protein kinases resides within a highly conserved kinase domain, which in eukaryotes ranges between 250 and 300 amino acids. The kinase domain catalyzes a phosphorylation reaction by binding a substrate to its active center, which transfers the phosphate in the gamma position of adenosine triphosphate (ATP) to the hydroxyl group of serine/threonine (serine/threonine kinases) or tyrosine (tyrosine kinases) side chains.

Most eukaryotic protein kinases can be grouped into seven superfamilies based on the conservation of functional amino acid motifs and substrate specificities [1,3,4]. The AGC protein kinase superfamily was first defined by Hanks and Hunter in 1995 as proteins containing kinase domains most similar to PKA, PKG and PKC [4,5]. The superfamily contains more than 10% of over 500 human kinases. A total of 42 of the AGC kinases are modular proteins containing additional protein domains that are involved in regulating the activity and localization of the kinase. The catalytic Ser/Thr kinase domain of AGC kinases is composed of two lobes, which are linked through a hinge region [3,6,7]. All 14 AGC kinases whose structure has been solved adopt this bi-lobal kinase fold, where the beta-strand-based N-lobe and the alpha-helical C-lobe sandwich one molecule of ATP. Interactions between a phosphorylated Ser in the C-terminal tail and positively charged residues in the N-lobe stabilize the active conformation of the kinase domain [7,8]. The N-lobe and the C-lobe surround the active site cleft, at which substrates interact with the kinase domain [9,10].

Phospho-regulation through AGC kinases is vital, as exemplified by many disease-associated mutations in genes encoding AGC kinases, pointing to the deregulation of these kinases in various types of cancer and diabetes, as well as other inherited disorders [5]. Members of the AGC kinase family respond to various extracellular signals and are involved in a range of well-characterized signaling pathways in which they phosphorylate distinct protein substrates. Although many AGC kinases have been the focus of molecular cell biology research to reveal their molecular mechanisms in diverse biological processes, some groups within the AGC kinase superfamily, including the microtubule-associated serine/threonine (MAST) kinases, remain poorly understood.

The first member of the MAST kinase family to be identified was MAST2 (originally named MAST205). MAST kinases are modular proteins containing three evolutionary conserved protein domains (Figure 1). The founding member, MAST2, was discovered as a protein associated with the spermatid manchette in mouse testes and has been suggested to play a role in sperm maturation [11,12]. Although the name suggests that MAST kinases are directly associated with microtubules, a direct interaction of MAST kinases with microtubules has not yet been demonstrated. In the case of the mouse spermatid, the interaction of MAST2 with microtubules depended on microtubule-associated proteins (MAPs) and required the kinase domain of MAST2, as well as a region comprising the PDZ domain [11]. In this article, we will review the current literature on this family of protein kinases with a special focus on structure and function, interacting proteins, involvement in human diseases and emerging developmental model systems.

## 2. Domain Composition of MAST Kinases

Members of the MAST kinase family are modular proteins characterized by a conserved domain composition and arrangement. The central Ser/Thr kinase domain is flanked towards the C-terminal end by a PDZ domain (post-synaptic density protein, disc large and Zonula occludens-1) and towards the N-terminus by a DUF (domain of unknown function) 1908 domain (Figure 1). The DUF1908 domain is a common feature of all MAST kinases; however, the function of this domain with respect to the activity of MAST kinases remains poorly understood. The DUF1908 domain (PFAM ID PF0826) is about 275 amino acids long, and within the human MAST kinases, this size differs only slightly. The predicted structure of the DUF1908 domain can be divided into an unstructured N-terminal half and a structured C-terminal half containing 8 alpha-helixes (Figure 2). A striking feature of the N-terminal domain of MAST kinases is the fact that their overall sequence is highly enriched serine, tyrosine and threonine residues, which are potential targets for posttranslational modifications including phosphorylation.

The PDZ domains, in contrast to DUF1908, are well studied and occur in proteins of a variety of organisms such as bacteria, yeast, plants, and metazoans, including *Caenorhabditis elegans*, *Drosophila melanogaster* and *Homo sapiens* [17]. PDZ domains are often found in multidomain proteins and contain 80–100 amino acids that are arranged in five beta strands and two alpha helices [18,19,20]. Within a given protein, single or several copies of PDZ domains can occur; MAST kinases contain only one PDZ domain in the carboxy-terminal half of the protein (Figure 3). PDZ domains are major hubs for protein-protein interactions, and their binding specificity is often based upon conserved amino acid motifs at the very carboxy-terminus of ligands [19]. PDZ domains can be divided into three classes based on their binding specificities. Class 1 PDZ domains recognize the binding motif X-S/T-X-V/I/L (where X represents any amino acid). MAST2-PDZ is an example of this class, as it recognizes this motif in its ligand PTEN [21]. Class 2 PDZ domains recognize a motif of four amino acids, alternating between a hydrophobic and a variable amino acid, and the third class recognizes the motif X-D/E-X-⌽ [22,23]. Another mode of binding allows the complexation of PDZ domains with each other, thus causing PDZ-based dimerization, which occurs mainly in proteins that have more than one PDZ domain [24,25]. Homodimer formation was also observed for proteins with only one PDZ domain; notably, this mode of binding was reported for MAST2-PDZ [21,22]. This self-association due to PDZ binding suggests an influence of this domain on the regulation of MAST kinases through dimerization.

The core domain of the MAST kinase family is a Ser/Thr kinase domain divided into two subdomains. Since the MAST kinases belong to the AGC kinase family, the amino acid sequences of the human MAST kinase domains show a high degree of conservation compared to AGC kinases, for example, from 34% to 36% sequence identity and 54% to 59% amino acid sequence similarity when comparing PKA, PKC and PKG (Figure 4). Moreover, the catalytic domain of the MAST kinases shares with the AGC kinases all highly conserved motifs such as DFG, APE and HRD, which are important for ATP binding and magnesium transfer and are therefore essential for kinase activity and activation [3,4,5]. However, there is one exception, which is not found in other AGC kinases: the first glycine within the conserved glycine-rich loop (GXGXXG), is replaced by a serine in MAST kinases [11]. This exchange of glycine for a serine allows speculation about possible phosphorylation at this site, which could influence the regulation of the kinase activity. The structure of the modulatory C-terminal tail region of MAST kinases is also well conserved (Figure 5). The comparison of the sequence of the C-terminal tail of the human MAST kinases with PKA and PKG revealed a 42–57% amino acid similarity and 29–43% identity. In summary, the overall structure of the MAST kinase domains exhibits a high degree of structural and sequence conservation with other members of the AGC kinase family.

## 3. Substrates and Interactors of MAST Kinases

A range of different MAST kinase-interacting proteins have been reported, some of which represent potential substrates of MAST kinases (Table 1). MAST1, 2 and 3 were all shown to bind to the lipid phosphatase PTEN (Phosphatase and Tensin homolog) [26,27]. PTEN is a key regulator of cell growth and cell survival, and as such, represents an important tumor suppressor in humans [28]. The interaction of PTEN with MAST2 was shown to facilitate the phosphorylation of PTEN in vitro [26]. Phosphorylation of the carboxy-terminal domain of PTEN reduces its activity and prevents PTEN from degradation, suggesting that MAST2 regulates PTEN activity and stability in vivo [29]. This MAST kinase-dependent inhibition of PTEN activity was also observed in multiple myeloma cell lines, where knockdown of MAST4 leads to reduced PTEN activity and increased activity of proteins involved in the mTOR signaling pathway [30]. The PTEN/MAST2 interaction might also play an important role in neuronal homeostasis. In human neuroblastoma cells, complex formation of PTEN and MAST2 was reported to suppress neurite outgrowth [31,32]. This mechanism also plays an important role during a rabies virus infection, where the glycoprotein (G protein) of the rabies virus (RABV) competes with PTEN to interact with MAST2 through its class 1 PDZ binding site [21,33].

Another MAST kinase binding partner and potential substrate is the Na^+^/H^+^ exchanger NHE3 [34]. NHE3 is a major Na^+^/H^+^ exchanger that functions in the apical membrane domain of renal and intestinal epithelial cells, in particular in the renal proximal tubules [35]. The interaction of MAST2 with NHE3 requires the PDZ domain of MAST2. Co-expression of MAST2 and NHE3 in opossum kidney cells caused an inhibition of pH recovery, suggesting that MAST2 inhibits NHE3’s ion transport function. The mechanism of this regulation might involve the phosphorylation of NHE3 by MAST2 [34].

The earliest studies on MAST kinases suggested that these proteins are part of large multi-protein complexes and that MAST2 presumably binds to MAPs [11]. In addition to PTEN and NHE3, many other proteins have been reported to interact with MAST kinases (Table 1). MAST1 has been found to bind β2-syntrophin, a component of the dystrophin/utrophin network within the plasma membrane cortex at neuromuscular synapses [36]. The interaction of MAST1 and β2-syntrophin involves the PDZ domains of both proteins, and it was proposed that MAST kinases might link the dystrophin complex via β2-syntrophin with the microtubule network. Additionally, protocadherin LKC (expressed in liver, kidney and colon tissues) (PCLKC) protein was reported to bind MAST2 through its PDZ domain [37]. The interaction of MAST2 with PCLKC resembles a canonical interaction between the PDZ domain of MAST2 and a carboxy-terminal PDZ-binding motif in PCLKC. Interestingly, this member of the atypical cadherin superfamily acts as a tumor suppressor by inducing contact inhibition in epithelial cells and is frequently lost in cancer cells [37].

Proteins involved in immune responses have been found to interact with MAST2, including TRAF6, a component of the NF-κB pathway [38]. The interaction with TRAF6 is dependent on the amino-terminal region of MAST2 and is presumed to function in the regulation of immune responses. In this context, it is interesting that the induction of proinflammatory cytokines, including interleukin 12 (Il12) by LPS involves MAST2 [38,39]. Xiong and colleagues (2004) proposed that the association of MAST2 with TRAF6 might result in the inhibition of TRAF6-dependent NF-κB activation. Signaling through NF-κB controls gene transcription in response to extracellular signals, which in turn affects important cellular functions, in particular during immune responses and inflammation, and its incorrect regulation has been linked to many human diseases (see below). A potential role for MAST3 in inflammatory bowel disease may be explained by the misregulation of the same pathway [40]. However, the details of how MAST2 affects the activity of TRAF6, or PCLKC function, remain to be resolved.

Analyses of the phosphoproteome of 14-3-3 revealed that MAST2 directly interacts with 14-3-3 proteins in a phosphorylation-dependent manner [41]. 14-3-3 proteins are ubiquitously expressed in eukaryotic organisms and display a plethora of interacting proteins that are involved in numerous biological processes [42]. There are 7 isoforms found in humans: β, γ, ε, ζ, η, θ and σ [43]. However, direct interaction between MAST2 and 14-3-3 has been demonstrated by Western assays using an antibody binding to the N terminus of 14-3-3 β, and it remains currently unclear whether the other isoforms also interact directly with MAST2.

**Table 1 ijms-24-11913-t001:** MAST kinase interacting proteins *.

MASTK	Interactor	Phosphosite ^1^	Interaction Domain ^2^	References
MAST1	USP1	-	PDZ	[44]
MAST1	Cdh1	-	PDZ	[44]
MAST1	CHIP	-	K317, K545	[45]
MAST1	MEK	S221	-	[46]
MAST1	c-Raf	-	-	[46]
MAST1	HSP90	-	-	[45]
MAST1 MAST2	SNTB2	-	PDZ	[36]
MAST1 MAST2 MAST3	PTEN	C-term.	PDZ	[26]
MAST1 MAST2	MAPs MAPs	-	KD + aa 948–1212	[11,47]
MAST2	TRAF6	-	N-term.	[38]
MAST2	RABV-G	-	PDZ	[21]
MAST2	CFTR	-	PDZ	[48]
MAST2	NHE3	n.d.	PDZ	[34]
MAST2	PCLKC	-	PDZ	[37]
MAST2	14-3-3	-	-	[41]
MAST3	ARPP-16	S46	-	[49]
MAST4	Sox9	S494	-	[50]

* The table summarizes published interactions of MAST kinases with other proteins. Where known, the phosphorylation site (Phosphosite) ^1^, if the interactor was identified as a substrate, and the interaction site ^2^ within MAST kinases are listed. An empty field (“-”) indicates that a phosphorylation site or interaction domain has not been described. MASTK = MAST kinase; KD = kinase domain; n.d. = not determined.

A well-confirmed substrate of MAST3 is ARPP-16 (cAMP-regulated phosphoprotein of molecular weight 16 kDa), an alternatively spliced variant of ARPP-19 [49]. ARPP-16 and ARPP-19 proteins represent cAMP-regulated phosphoproteins, as they are substrates of PKA and highly expressed in the medium spiny neurons in the striatum of the brain [51,52,53]. MAST3 is part of a complex in which it competes with PKA for the ARPP16-dependent regulation of protein phosphatase 2A (PP2A) in the striatum. PP2A activity is inhibited by the phosphorylation of serine 46 of ARPP-16 by MAST3 [49,54]. In contrast, the phosphorylation of serine 88 in ARPP-16 by PKA leads to a decreased phosphorylation of serine 46 and prevents the inhibition of PP2A [52]. Moreover, MAST3 is regulated by PKA through phosphorylation at threonine 389 [54]. In summary, these results demonstrate the important role of MAST3 in the regulation of PP2A in neuronal cells.

MAST kinases have also been reported to interfere with transcriptional regulation. Sox9 belongs to the family of high-mobility group domain transcription factors and is an important regulator in the differentiation of mesenchymal stromal cells into chondrocytes [55,56,57]. In this system, MAST4 acts as a negative regulator, as by phosphorylating Sox9 at serine 494, Sox9 is targeted for degradation through the proteasome. This MAST4-dependent degradation of Sox9 suggests an essential role for MAST4 in the differentiation of mesenchymal stromal cells [50]. The ETS transcription factor family member ERM (Ets-related molecule) is one essential component of the gene regulatory network in spermatogonial stem cell self-renewal, and it is controlled by the fibroblast growth factor (FGF) 2. Phosphorylation of ERM by MAST4 at serine 367 enhances the transcription of ERM-target genes, suggesting that MAST4, together with the FGF2 signaling pathway, is involved in spermatogonial stem cell self-renewal by regulating the transcription factor ERM [58].

## 4. MAST Kinases in Human Disease

As discussed in the previous section, members of the MAST kinase family interact with a large variety of distinct binding partners, which may act as potential regulators or substrates. In mice, the genes encoding MAST kinases are widely expressed across various tissues such as the brain, heart, spleen, lung, liver, skeletal muscle, kidney and testis, suggesting a large diversity of functions [59]. Consistent with their broad expression patterns, malfunctions of MAST kinases are implicated in a wide range of human diseases affecting distinct tissues and organs, including breast cancer, inflammatory bowel disease, neuronal disorders and cystic fibrosis (Table 2) [21,31,33,34,40,48,60,61,62,63].

### 4.1. Association of MAST Kinase Mutations in Cancer

Somatic alterations in the genes for MAST1 and MAST2 were discovered and characterized in material obtained from breast cancer samples [60,64]. An increased risk for breast cancer was also reported to be associated with nine SNPs (single nucleotide polymorphism) found in MAST2 [61]. On the molecular level, three types of gene fusions were found for the MAST1 gene (with ZNF700, with taDA2A, and with NFIX) and two types were discovered in the case of the MAST2 gene (with ARIDA1a and GPBP1L1) in different transcriptomes from breast cancer samples and breast cancer-derived cell lines [60]. Overexpression of these fused gene products lead to enhanced proliferation in a benign breast cell line and RNAi-mediated knockdown of MAST2 in a cancer cell line with MAST2 gene fusions led to reduced growth and reduced tumor formation in mice xenografts. Together, these studies implicate the MAST1 and MAST2 gene fusions in the development of invasive breast carcinomas.

Alterations in MAST kinase gene expression have been revealed as a major feature of the relationship of this family of protein kinases with cancer. MAST2 was identified in a screen for pro-survival factors in a cDNA library of a glioblastoma cell line. Further studies showed that MAST2 overexpression promotes cell survival and cell growth, and the knock-down of MAST2 led to reduced tumor growth in a xenograft tumor mouse model, supporting a role of MAST2 in tumor progression that is characteristic of an oncogene [65]. Altered RNA levels of MAST kinases are associated with liver cancers and circular MAST1 RNA is upregulated in hepatocellular carcinoma and may serve as a diagnostic marker [66]. MAST2 mRNA is upregulated in liver cancer and expands the diagnostic spectrum of the disease [67]. MAST1, MAST2 and MAST3 were all found to bind to the crucial tumor suppressor Adenomatous polyposis coli (APC); however, the functional significance of this interaction has yet to be investigated [68].

MAST4 was identified as an estrogen response gene that was upregulated in female patients with multiple myeloma (MM) associated with low osteolytic lesions [69]. A recent investigation confirmed this association and revealed a mechanistic insight into the role of MAST4 in MM [30]. MM can be accompanied by characteristic lytic bone lesions derived from malignant plasma cells accumulating in the bone marrow, a syndrome called MM bone disease (MMBD). Estrogen levels are correlated with the extent of the disease, raising the hypothesis that estrogen may enhance the severity of myeloma. The search for estrogen response genes revealed, among others, MAST4. The study further showed that MAST4 is a negative regulator of MMBD and that the levels of MAST4 in MM cells negatively correlate with the severity of MMBD; high estrogen pathway activation levels correlated with high levels of MAST4 [30]. Pathway analyses suggested that MAST4 regulates the PI3-Kinase and mTOR pathways and that MAST4 colocalizes and co-immunoprecipitates with PTEN in MM cells. MAST4 knockdown resulted in an upregulation of Pi3-Kinase and mTOR signaling. Furthermore, the MAST4 gene harbors an estrogen receptor-responsive element in the enhancer of the gene, providing further evidence that MAST4 is indeed a target of estrogen receptor transcriptional activity.

Since the MAST kinases are known to be associated with different cancer types, further studies addressed the biological role of these kinases and identified MAST1 as a key regulator for cisplatin resistance in human cancer cells. Cisplatin is one of the most commonly used chemotherapeutic agents, as it blocks cell proliferation by inhibiting DNA replication [70]. A major complication of cisplatin chemotherapy is the upregulation of the ERK mitogen-activated protein kinase (MAPK) cascade, which contributes to cisplatin resistance by inhibiting cisplatin-induced apoptosis [71]. MAST1 was found to form a complex with components of the ERK pathway, including MEK1 and its upstream kinase, cRaf. Cisplatin interferes with this complex formation and causes the dissociation of cRaf but not MAST1, which then activates MEK1 by phosphorylation at serine 221 [46,72]. The regulation of MAST1 is based on the heat shock protein 90 (hsp90) and the E3 ubiquitin ligase CHIP. The direct binding between MAST1 and hsp90 increases the stabilization of MAST1, as the ubiquitination of lysines 317 and 545 of MAST1 by CHIP is inhibited [45]. A recent study revealed that the stability of MAST1 is also enhanced by the deubiquitinase USP1 in cisplatin-resistant cancer cells [44]. Together, these findings suggest that Hsp90, MAST1 and USP1 may be excellent targets for cisplatin-resistant cancer cells, as their inhibition will enhance cisplatin sensitivity and result in increased cell death. The negative side effects of cisplatin-based chemotherapy are also associated with increased inflammatory responses, which can be ameliorated by glucosteroid receptor (GR) agonists. A recent study found that MAST1 transcription is upregulated through GR activation and thus mediates resistance through the aforementioned reprogramming of the ERK pathway [73].

### 4.2. MAST Kinases in Neuronal Diseases

Rabies infections are deleterious viral infections of the brain. MAST1 and MAST2 were shown to interact with the glycoprotein of the virulent rabies virus in infected cells, and the binding was shown to inhibit normal MAST2 localization to apical membranes [33,63]. An interaction between the PDZ domain of MAST2 and the C-terminal residues of the glycoprotein of the rabies virus was required for the survival of infected neuronal cells, a signature of virulence [33,63]. Rabies virus glycoprotein was reported to disrupt interaction between PTEN and MAST2 [21]. PTEN was shown to bind to the PDZ domains of MAST1, MAST2 and MAST3, and phosphorylation of PTEN in vitro by MAST2 indicates that PTEN may be a substrate of MAST kinases [26]. The PDZ domain of MAST2 binds both PTEN and glycoprotein with similar affinity, and the viral glycoprotein disrupts the interaction of MAST2 with PTEN. The presence of glycoprotein decreased the nuclear localization of PTEN, which results in increased neuronal survival after infection with the rabies virus [21]. Therefore, the MAST kinases may be an important regulator of PTEN localization and activity in neuronal cells.

The aforementioned interaction between MAST2 and PTEN inhibits PTEN function by preventing neuronal outgrowth and regeneration. This mechanism might explain the enhanced neuronal survival following rabies virus infection. In support of this idea, an RNAi screen designed to identify protein kinases that regulate growth cone collapse, neurite retraction and neurite outgrowth showed that RNAi knockdown of MAST2 promotes neurite outgrowth and inhibits (lipopolysaccharide (LPS)-induced) neurite retraction in rat primary midbrain neurons [31]. Therefore, an overall function of MAST2 might be to control PTEN localization and activity in the nervous system, suggesting that the MAST kinases are involved in regulating processes that are often disrupted in neuronal disorders.

Both MAST1 and MAST2 have been found to interact with β2-syntrophin, coupling MAST2 via its PDZ domain to the dystrophin/utrophin network at the neuromuscular junction (NMJ) [36]. The NMJ and neuronal postsynaptic densities are composed of receptors, receptor clustering elements, cytoskeletal components and signal transduction molecules. The interaction between β2-syntrophin and MAST2 may function to link the dystrophin and utrophin networks to cellular signaling pathways.

### 4.3. MAST Kinases in Cystic Fibrosis and Diarrhea

MAST kinases are also linked to cystic fibrosis and secretory diarrhea. MAST2 was shown to form a complex with the key factor associated with cystic fibrosis, the cystic fibrosis transmembrane conductance regulator (CFTR) [48]. CFTR represents the arguably best-known anion channel specific for the transport of chloride (Cl^–^) and bicarbonate (HCO_3_^–^) through the apical membranes in most, if not all, epithelia-producing mucus, including the lung and intestine. CFTR interacts with several proteins, including PDZ domain proteins, which CFTR binds through its conserved C-terminal PDZ-binding motif [74,75,76,77,78]. The PDZ-based interactions of CFTR with PDZ-domain proteins play important roles in the biosynthesis and membrane transport of the protein [79,80]. The Golgi-associated PDZ protein CFTR-associated ligand (CAL) regulates the cell surface levels of CFTR and can promote lysosomal degradation of CFTR [81,82]. The binding between CFTR and CAL is sensitive to competitive binding to other PDZ domain proteins, including MAST2; indeed, high levels of MAST2 expression have been shown to increase CFTR surface levels, while knock-down of MAST2 inhibits CFTR function [48].

Besides its functional importance in cystic fibrosis, CFTR also represents a major chloride and hydrogen carbonate ion channel in the intestinal mucosa to control ion and pH homeostasis in the gut [78]. In addition, MAST2 also binds to and phosphorylates another important intestinal ion channel, the Na+/H+ exchanger NHE3 [34]. The phosphorylation by MAST2 was shown to inhibit NHE3 activity. NHE3 is important to maintain normal gastrointestinal physiology and its malfunction leads to impaired absorption and can increase the fluidity of diarrhea [48]. The results, which show that MAST2 regulates NHE3 as well as CFTR function suggest MAST2 as a potential cross-regulator that can assemble both channels and maybe other proteins into a macromolecular complex, thereby increasing its impact on intestinal physiology.

### 4.4. MAST Kinases in Male Fertility

Since MAST2 was the first of the MAST kinases to be discovered in the testis in mice, it was not surprising that MAST2 was reported to be associated with infertility in humans [11,12]. A study that suggested recurrent copy number variations as the cause of idiopathic nonobstructive azoospermia showed that, among other copy number variations, the presence of duplication of MAST2 is a risk factor for this severe form of male infertility [83]. Besides MAST2, MAST4 is associated with spermatogenesis in mice as it controls the self-renewal of spermatogonial stem cells by phosphorylating the transcription factor ERM. Knockout of MAST4 results in a reduced sperm number and the typical phenotype of SCO (Sertoli cell-only syndrome), which represents a form of male infertility [58].

### 4.5. MAST Kinases in Inflammation

Both MAST2 and MAST3 were associated with mechanisms that are required for NF-κB activity. MAST3 was found to be a factor in inflammatory bowel disease (IBD) tissues to increase Toll-like receptor (TLR) 4-dependent NF-κB activity and knock-down of MAST3 resulted in reduced NF-κB activity [40]. A follow-up study reported that MAST3 acts on the NF-κB pathway by changing the expression of several genes in the gut of IBD patients and might thereby trigger immune reactions [62]. However, the direct target of MAST3 in this process remains to be determined. Nevertheless, these studies highlight the potential involvement of MAST3 in the pathogenesis of IBD.

In B-cells, the expression of MAST2 was shown to be under the control of the class II MHCII transactivator, a key regulator of the MHCII response [84]. Furthermore, MAST2 regulates LPS-induced NF-κB regulation by forming a complex with TRAF6 (TNF receptor-associated factor 6), which results in NF-κB inhibition and a reduction of inflammatory responses [38,39]. Thus, MAST2 and MAST3 might have opposite effects on NF-κB regulation depending on different stimuli: MAST2 exhibits an inhibitory influence in LPS-dependent NF-κB regulation, while MAST3 may be enhancing NF-κB target activation in TLR4-dependent NF-κB regulation.

**Table 2 ijms-24-11913-t002:** Overview of MAST Kinase-associated human diseases **.

Disease Subgroup ^1^	MASTK	Disease ^2^	Cause ^3^	References
**Infertility**	MAST2	Nonobstructive azoospermia	Gene Duplication	[83]
**Cancer**	MAST4	Multiple myeloma bone disease	Overexpressed	[30]
		Acral melanoma	Various deleterious mutations	[85]
		Ductal carcinoma in-situ Invasive breast cancer	Upregulated	[86]
	MAST3	Prostate cancer	Gene conversion	[87]
	MAST2	Cutaneous melanoma	Translocation	[88]
	MAST2	Esophageal cancer Pancreatic cancer Sarcomas	Overexpressed	[65]
	MAST2	Liver cancer	Overexpression	[67]
	MAST2	Chronic myeloid leukemia	Insertion of exon 8 in a BCR-ABL1 fusion gene	[89]
	MAST2	Breast cancer	Translocation	[64]
	MAST2	Breast cancer	Gene fusion	[60]
	MAST1			
	MAST1	Breast cancer	High levels of DNA methylation	[90]
	MAST1	Non-small-cell lung cancer	Upregulated	[91]
	MAST1	Pheochromocytoma, paraganglioma	Overexpression by hypomethylation	[92]
	MAST1	Hepatocellular carcinoma	Upregulated circRNA	[66]
	MAST1	Uterine corpus endometrial carcinoma	Upregulated	[93]
	MAST1	Lung cancer	S81Y C291F V316E	[94]
**Cardiovascular** **Diseases**	MAST2	Venous thrombosis	R89Q	[95]
**Neuronal diseases**	MAST3	Developmental and epileptic encephalopathy	S101F S104L G515S L516P	[96]
	MAST3	Developmental and epileptic encephalopathy	G510S G515S	[97]
	MAST4	Juvenile myoclonic epilepsy (JME)	T347M	[98]
		Childhood absence epilepsy	P1201R	
	MAST1	Intellectual disability, speech delay, hypotonia, facial dysmorphism, autism	S93L	[99]
	MAST1	Cerebral palsy, intellectual disability	P500L	[100]
	MAST1	Intellectual disability	P1177R	[101]
	MAST1	Neurologic abnormalities, developmental disability, mental retardation	Deletion	[102]
	MAST1	Intellectual disability	L1180R	[103]
	MAST1	Congenital bilateral Perisylvian syndrome	Deletion of Q223 to D230	[104]
	MAST1	Mega-corpus-callosum syndrome with cortical malformations without cerebellar Hypoplasia	G522E	[105]
	MAST1	Mega-corpus-callosum syndrome with cerebellar hypoplasia and cortical malformations	L278del E194del K276del G517S E697del	[47]
**Inflammatory bowel disease**	MAST3	Crohn’s disease (CD) and ulcerative colitis (UC)	S861G	[40]
**Others**	MAST4	Asthma (horses)	Overexpressed	[106]
	MAST3	Hepatic steatosis	Intronic variant	[107]
	MAST3	Rheumatoid arthritis	Overexpressed in fibroblast-like synovial cells	[108]
	MAST2	Type 2 diabetes mellitus	A1463T	[39]
	MAST2	Rabies infection	Viral glycoprotein prevents complex formation (MAST2-PDZ and PTEN) and promotes neuronal survival	[21,33,63,109,110]

** The table summarizes human diseases associated with MAST kinases (MASTK). Diseases were assembled into subgroups ^1^ and the exact disease ^2^ name and its cause ^3^ are indicated.

## 5. MAST Kinases in Model Organisms

The human MAST kinases exhibit a remarkably conserved modular domain composition and arrangement (Figure 1). Using BLAST search analyses, homologs of MAST kinases were found in organisms of the metazoan kingdom. BLAST searches against annotated proteins demonstrate a high degree of amino acid sequence conservation of the MAST kinase domain in vertebrates, insects, nematodes and simple metazoans (Figure 6). All MAST kinase homologs identified in this survey encode proteins with one DUF1908, one Ser/Thr kinase domain and one PDZ domain as identified using the domain detection tool SMART [111,112]. Interestingly, in both exemplar nematodes (*Caenorhabditis elegans*, *Ascaris suum*) the DUF1908 domain is split into two subdomains. This split of the DUF1908 domain was not found in any other class than nematodes. It is also noticeable that due to evolutionary gene duplication, higher-evolved species such as humans, mice, and frogs possess four MAST kinases, while simpler species such as insects, nematodes, and simple metazoans comprise only one MAST kinase. Since the overall domain composition and structure are highly conserved and the human genome encodes four MAST kinases, model organisms provide simple and genetically tractable experimental systems to investigate the molecular mechanisms of MAST kinase function in vivo.

Mice serve as excellent model systems to examine the effect of human disease-related mutations within genes encoding human MAST kinases and to analyze their impact on the activity, regulation and localization of the kinase. MAST kinases are broadly expressed in the central nervous system [59]. Mutations in human MAST1 are associated with the mega-corpus-callosum syndrome, which includes cerebellar hypoplasia and cortical malformations in humans. One of these mutations, deletion of Lysine 278 (L278del), was introduced into a transgenic mouse line [47]. In heterozygous animals, an increase in the apoptosis of neurons and an enlarged corpus callosum were observed. The L278del in MAST1 was associated with reduced protein levels of the MAST2 and MAST3 kinases, suggesting that the regulation of MAST kinases may be regulated by interdependent mechanisms. Interestingly, the deletion of the MAST1 gene in mice is viable, suggesting that MAST1 either is not essential or the other members of the MAST kinases may compensate for the lack of MAST1 [47]. In support of the latter idea, it was reported that the individual deletion of either MAST kinase gene is not lethal but rather leads to distinct phenotypes. For example, the knockdown of MAST1 results in a decreased heart weight; the knockout of MAST2 causes an increased bone density and bone mineral content, and mice where MAST3 is deleted display behavioral problems resulting in hyperactivity as well as developing a cataract and increased circulating alkaline phosphatase level [116]. Mice with a truncated MAST4 kinase (E726stop) show a craniofacial phenotype caused by dental malocclusions [117] and complete knockdown of MAST 4 is associated with an osteoporosis phenotype [50]. In addition, the knockdown of MAST4 reveals its role in spermatogenesis as it leads to decreased testes size and sperm number and the typical phenotype of Sertoli cell-only syndrome (SCO) [58].

In simpler model organisms such as nematodes (*Caenorhabditis elegans*) or insects (*Drosophila melanogaster*), the MAST kinases are expressed by a single gene. This is an advantage, as mutations cannot be compensated for by other MAST kinase family members in the genome. In *C. elegans*, the MAST kinase homolog KIN-4 is involved in the control of thermotaxis in conjunction with the stomatin homolog MEC-2 [118] and the diacylglycerol kinase DGK-1 [119,120]. In this system, the roles of KIN-4, DGK-1 and MEC-2 regulate presynaptic mechanisms, which control the release of neurotransmitters at the synapse of thermosensory neurons [121]. The *C. elegans* Kin-4 has also been shown to be required for longevity through a mechanism involving binding of PTEN and its contribution to the insulin/IGF-1 signaling pathway in the regulation of lifespan [122]. It is interesting to note that PTEN is also a known interaction partner of human MAST kinases; however, an association between human MAST kinases and the regulation of lifespan is yet unknown [26].

In the genetic model organism *Drosophila melanogaster*, a single MAST kinase homolog was identified as a recessive female sterile mutation called *drop out* (*dop*) [123,124]. Females homozygous for the mutant *dop*^1^ allele lay eggs, but the embryos exhibit severe morphogenetic defects early in embryogenesis. The developmental stage at which *dop* mutant embryos become abnormal is called cellularization, a process that transforms the syncytial blastoderm into a polarized blastoderm epithelium [125,126]. The phenotype of *dop* mutant embryos includes severely reduced membrane growth and a dropping of the nuclei out of the cortical cytoplasm, and most embryos fail to undergo gastrulation and further development [124]. The analyses of the cell biological role of *dop* in cellularization revealed that the membrane growth defect is associated with the mislocalization of several membrane proteins that are essential for the assembly of membrane cortex domains during cellularization [126,127]. These proteins include the apical scaffold protein and Par-3 homolog Bazooka (Baz) and the adherens junction protein E-cadherin [123]. Another membrane subdomain, the furrow channel marked by Slam, the small GTPase Rho1 and the scaffold protein dPatJ exhibited an overlap with a lateral membrane marker Discs large (Dlg) suggesting a requirement of Dop for compartmentalization of the plasma-membrane associated cortex [123]. Interestingly, all known phenotypes of *dop* mutants share the common feature that the defects can be linked to impaired dynein-dependent transport along microtubules. This includes the apical transport of lipid droplets in the embryo, the apical transport of ribonucleoprotein particles, the apical transport of Baz and the apical transport of Golgi-derived membrane reservoirs that fuel membrane growth in cellularization [123,128,129,130]. Genetic and biochemical studies revealed that Dop possibly regulates the dynein complex by phosphorylating the dynein intermediate chain (Dic) [123]. A domain function analysis in flies suggests that the DUF1908 and the PDZ domains might have regulatory functions, and further studies should reveal how the kinase activity of the fly MAST kinase homolog is controlled in embryogenesis, which may reveal important general properties of MAST kinase regulation in general.

## 6. Conclusions and Outlook

MAST kinases are involved in an increasing number of diverse biological processes ranging from acute and chronic human diseases to stem cell maintenance and longevity. The modular structure of MAST kinases enables interaction with many distinct proteins, facilitating substrate recognition and binding and bearing the potential for regulatory inputs. In several instances, competitive interactions appear as a theme in the pathophysiological aspects of MAST kinases. Within the past 10 years, the first defined molecular mechanisms that link MAST kinase function to a particular protein substrate phosphorylation event that causes specific biological responses have emerged. More detailed mechanistic studies in human diseases, including the use of model organisms, are likely to reveal MAST kinase interactors and substrates that play crucial roles in various signaling pathways regulating distinct cellular functionalities.

## Figures and Tables

**Figure 1 ijms-24-11913-f001:**
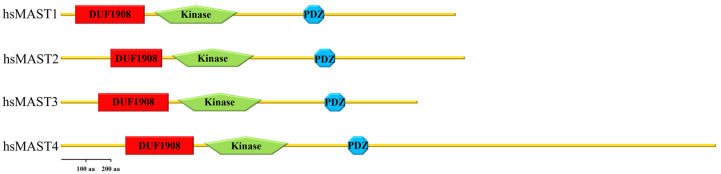
**Domain arrangement of human MAST kinases.** This cartoon demonstrates the domain composition and arrangement of the four family members of MAST kinases in humans. From N- to C-terminus, the DUF 1908 domain is followed by the serine/threonine kinase domain (drawn without its C-terminal extension) and the PDZ domain in the C-terminal half of the protein.

**Figure 2 ijms-24-11913-f002:**
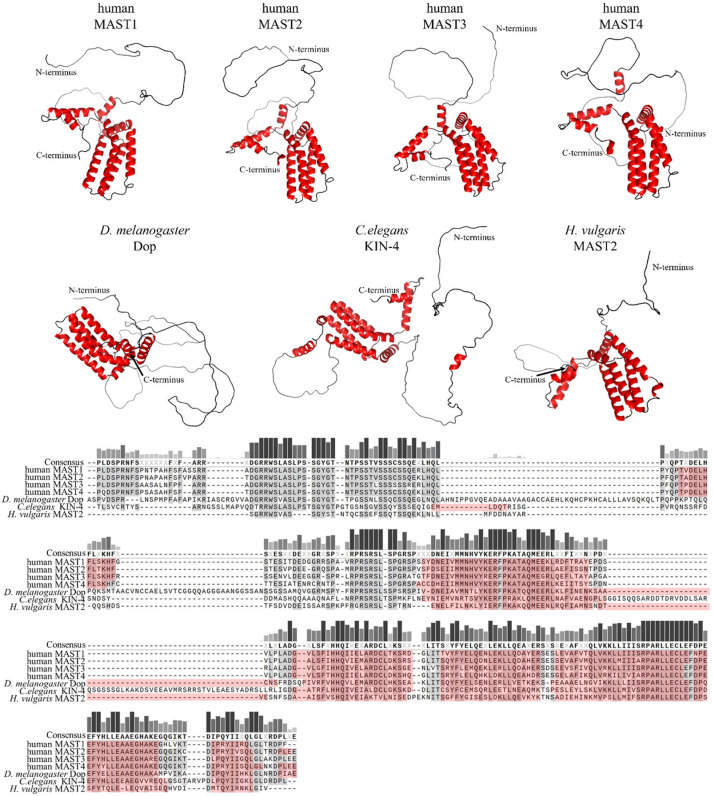
**Primary sequence comparison and structural comparison of DUF1908 domains in selected MAST kinases.** Top panel: AlphaFold predicted 3D models of DUF1908 of MAST kinases in humans, *Drosophila* (*D.*) *melanogaster*, *Caenorhabditis* (*C.*) *elegans* and *Hydra* (*H.*) *vulgaris* [13,14,15]. Alpha helices are shown in red. The 3D structures were oriented and labeled using the PyMol software (Schrödinger, L.; DeLano, W. PyMOL [Internet]. 2020. Available online: http://www.pymol.org/pymol; accessed on 5 December 2022). Lower panel: Multiple amino acid sequence alignment of DUF1908 of MAST kinases in different species using MAFFT (Snapgen; GSL Biotec, San Diego, CA, USA). The helices are marked in red. Conserved amino acids are highlighted in gray and dark red, respectively. A threshold of 55% was chosen for consensus. The bars above the consensus sequence indicate the degree of conservation; the darker and higher the bar, the higher the conservation. The sequences and domain annotations of the proteins were obtained from InterPro [16].

**Figure 3 ijms-24-11913-f003:**
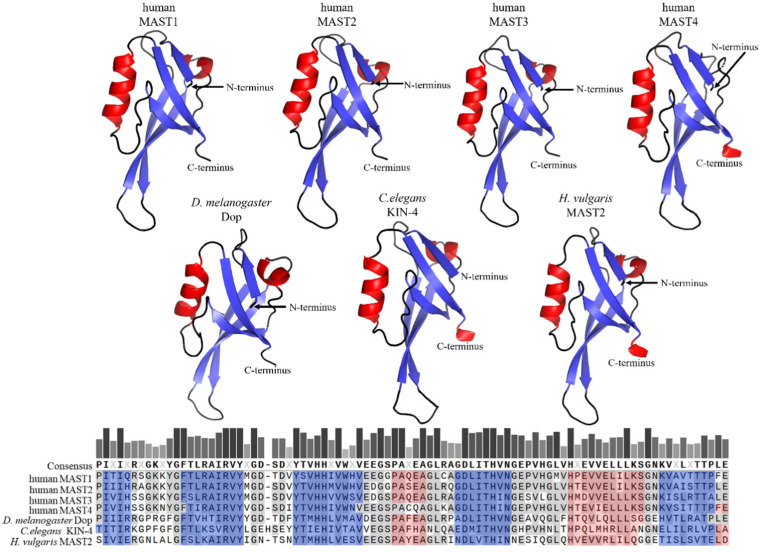
**Primary sequence comparison and structural comparison of PDZ domains in selected MAST kinases.** AlphaFold-predicted 3D models of PDZ domains of MAST kinases in humans, *Drosophila* (*D.*) *melanogaster*, *Caenorhabditis* (*C.*) *elegans*, and *Hydra* (*H.*) *vulgaris* [13,14,15]. Alpha helices are shown in red, and beta sheets in blue color. The 3D structures were oriented and labeled using the PyMol software (Schrödinger, LLC, New York, NY, USA). The structure of all four human MAST kinases was also solved experimentally (MAST1: https://doi.org/10.2210/pdb3ps4/pdb; MAST2: https://doi.org/10.2210/pdb2kyl/pdb; MAST3: https://doi.org/10.2210/pdb3khf/pdb; MAST4: https://doi.org/10.2210/pdb2w7r/pdb) (accessed on 14 July 2023). The structural models based on AlphaFold and based on experimental data do not exhibit appreciable differences. Lower panel: Multiple protein amino acid sequence alignment of PDZ domains of MAST kinases in different species using MAFFT (Snapgen; GSL Biotec). For further detail see the legend of Figure 2.

**Figure 4 ijms-24-11913-f004:**
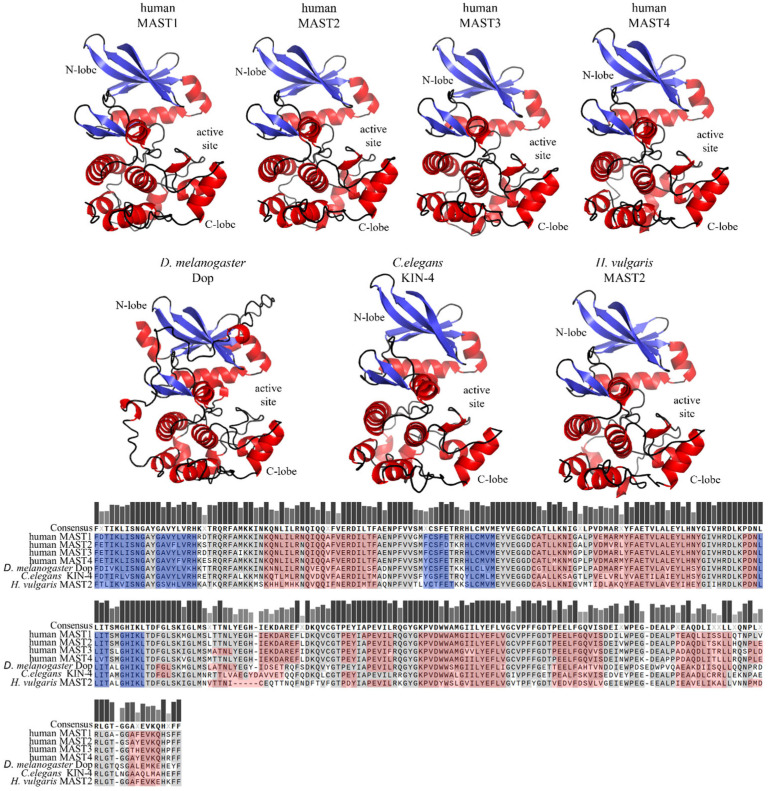
**Structure models and sequence comparison of the MAST kinase domains in different species.** Top panels: AlphaFold-prediction models of the kinase domain structure in humans, *Drosophila* (*D.*) *melanogaster*, *Caenorhabditis* (*C.*) *elegans*, and *Hydra* (*H.*) *vulgaris* [13,14,15]. The alpha helices are shown in red, and beta sheets in blue. The 3D structures were oriented and labeled using PyMol (Schrödinger, LLC). Lower panel: Multiple protein amino acid sequence alignment of MAST kinase domains of different species. Alignment was performed using MAFFT of the Snapgen software (GSL Biotec). For further detail see the legend of Figure 2.

**Figure 5 ijms-24-11913-f005:**
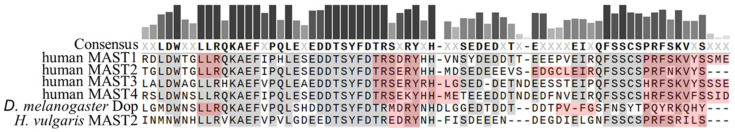
**Sequence comparison of the AGC-kinase C-terminal region in MAST kinases of different species.** Multiple protein amino acid sequence alignments of AGC-kinase C-terminal regions of MAST kinases in humans, *Drosophila* (*D.*) *melanogaster* and *Hydra* (*H.*) *vulgaris.* Alignment was performed using MAFFT (Snapgene; GSL Biotec). Predicted alpha-helical domains are marked in red and conserved amino acids are marked in grey or dark red.

**Figure 6 ijms-24-11913-f006:**
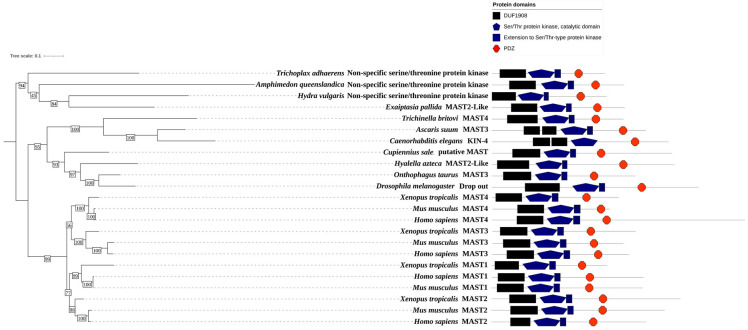
**Phylogenetic analyses of MAST kinases.** Maximum likelihood phylogenetic tree showing the relation of MAST, MAST-like and putative MAST kinases with similar protein domain architecture from simple organisms like *Trichoplax adhaerens*, *Hydra vulgaris* to higher organisms such as *Homo sapiens* and *Mus musculus*. The protein domain annotations were identified by SMART [113] and the alignment was performed by using MAFFT version 7 [114] and RAxML was used for phylogenetic analysis [115]. All bootstrap values are shown, and the tree image was drawn using the iTOL web server [111,112].

## Data Availability

Not applicable.

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
