# Peer review of "Microtubule-Associated Serine/Threonine (MAST) Kinases in Development and Disease"

_ijms, 2023, doi:10.3390/ijms241511913_

Round 1

Reviewer 1 Report

The review article “Microtubule-associated Serine/Threonine (MAST) Kinases in Development and Disease” by Rumpf et. al. is well written. However I have few concerns regarding the manuscript.

1.    Line 45 – “b-strand” should be written with symbol for beta and “a-helical” should have symbol for alpha.  

2.    The scientific names of organisms should be italicized.

3. Fig. 3 - While there is no appreciable difference in the alphafold prediction and experimentally determined structure of PDZ domains, experimentally determined structure should be preferred over structure prediction wherever possible. only when experimentally determined structure is not available, should a prediction be used. At-least the PDZ domains of all the human MAST kinases are experimentally determined so figure 3 should show the experimentally determined structure of PDZ domains instead of structure prediction.

4. Fig. 5 -  These models and information are misleading. First there are very little interactions in the models in 3 dimension and the models looks more like unstructured proteins with some secondary structures. These are models, not experimentally determined structures and even alpha fold has low confidence in this region of the models. I suggest this to be replaced by secondary structure prediction at best.

Author Response

We thank the referee for  their positive evaluation of our manuscript. In the following we would like to respond to the specific comments:

  1. Line 45 – “b-strand” should be written with symbol for beta and “a-helical” should have symbol for alpha.                                                             Response: We changed the writing accordingly and used 'alpha' and 'beta' to be consistent with the spelling in the entire manuscript.
  2. The scientific names of organisms should be italicized.                  Response: We Italicized all names of organisms in the text, figure legends and figures.
  3. Fig. 3 - While there is no appreciable difference in the alphafold prediction and experimentally determined structure of PDZ domains, experimentally determined structure should be preferred over structure prediction wherever possible. only when experimentally determined structure is not available, should a prediction be used. At-least the PDZ domains of all the human MAST kinases are experimentally determined so figure 3 should show the experimentally determined structure of PDZ domains instead of structure prediction.                                                                                                            Response: We appreciate and fully understand the point made by the referee about the advantage of experimental structural data rather than in silico modelling. However, structure prediction by AlphaFold has been widely applied and is accepted as a high fidelity structure prediction tool. As the referee points out, in the case of MAST kinase PDZ domains, the AlphaFold structure prediction does not differ from the model based on experimental. For reasons of consistency, we would therefore like to keep the AlphaFold predicted structural models in the manuscript. However to consider the referee's point, we have extended the figure legend to refer to the experimentally based models and to point out that there is not appreciable difference in the models based on experimental data or AlphaFold based predictions.  
  4. Fig. 5 -  These models and information are misleading. First there are very little interactions in the models in 3 dimension and the models looks more like unstructured proteins with some secondary structures. These are models, not experimentally determined structures and even alpha fold has low confidence in this region of the models. I suggest this to be replaced by secondary structure prediction at best.                                                         Response:  We agree with the referee. We decided to remove the AlphaFold predictions from Fig. 5 and only kept the sequence comparison with the prediction of secondary structures as the referee suggested.            

Reviewer 2 Report

The manuscript by Rumpf et al. presents a review on the role of Microtubule-Associated Serine/Threonine (MAST) kinases in protein phosphorylation and cell signaling. A relevant, up-to-date, and representative pool of publications was analyzed to elucidate the structure and biochemistry of MAST kinases. The authors of the review systematically describe the structure, different types of substrates, and the involvement of MAST kinases in human diseases. The review is important and deserve publication.

Specific comments:

Figure 2: In the figure caption, dark blue color is mentioned for the conserved amino acids. However there are no dark blue elements in the picture. The same refers to Figure 5 (blue color indicated in the caption is not present in the picture).

Table 1:  The footnote of the table is not referenced from the table itself. The same refers to Table 2.

I recommend acceptance of the manuscript for publication after minor corrections.

Author Response

We thank the referee for their enthusiastic comments and would like to respond on the their comments in the following. 

Figure 2: In the figure caption, dark blue color is mentioned for the conserved amino acids. However there are no dark blue elements in the picture. The same refers to Figure 5 (blue color indicated in the caption is not present in the picture).

Response: We thank the referee for noticing this mistake, which must have been a remnant of an earlier draft of the manuscript. We have corrected the figure legends accordingly.

Table 1:  The footnote of the table is not referenced from the table itself. The same refers to Table 2.

Response: We appreciate the detection of this omission. We have added a reference in each table that links the footnote to the table unequivocally.

Reviewer 3 Report

The review entitled ‘Microtubule-associated Serine/Threonine (MAST) Kinases in Development and Disease.’ is well written. The review has a logical progression from molecular biology to model organisms in describing the structure, function and role of these kinases in development and disease. This thorough review should be of interest to researchers specifically engaged in MAST kinase research as well as those involved in kinase inhibitor drug discovery. The review is well-referenced and this reviewer does not detect any glaring omissions.

There are a few typographical issues.

Line 45: α and β should be inserted for a- and b-.

Line 55: delete ‘in’.

Line 57: microtubule need not be capitalized.

Line 94: helices should not be capitalized.

A thorough check for proper capitalization use should be performed. Capitalization should not be used for generic drug names (cisplatin, line 293 and other places), chemical or protein names (stomatin and diacylglycerol, line 458 and dynein, line 483).

The review entitled ‘Microtubule-associated Serine/Threonine (MAST) Kinases in Development and Disease.’ is well written. The review has a logical progression from molecular biology to model organisms in describing the structure, function and role of these kinases in development and disease. This thorough review should be of interest to researchers specifically engaged in MAST kinase research as well as those involved in kinase inhibitor drug discovery. The review is well-referenced and this reviewer does not detect any glaring omissions.

There are a few typographical issues.

Line 45: α and β should be inserted for a- and b-.

Line 55: delete ‘in’.

Line 57: microtubule need not be capitalized.

Line 94: helices should not be capitalized.

A thorough check for proper capitalization use should be performed. Capitalization should not be used for generic drug names (cisplatin, line 293 and other places), chemical or protein names (stomatin and diacylglycerol, line 458 and dynein, line 483).

Author Response

We thank the referee for their positive evaluation of our manuscript. We would like to respond to the specific comments as follows.

There are a few typographical issues.

Line 45: α and β should be inserted for a- and b-.

Response: for reasons of consistency within the manuscript we have used 'alpha' and 'beta'. 

Line 55: delete ‘in’.

Response: We deleted 'in'

Line 57: microtubule need not be capitalized.

Line 94: helices should not be capitalized.

A thorough check for proper capitalization use should be performed. Capitalization should not be used for generic drug names (cisplatin, line 293 and other places), chemical or protein names (stomatin and diacylglycerol, line 458 and dynein, line 483).

Response: We corrected the capitalisations through out manuscript including the figure legends and tables.